# Statistical characteristics of convective wind gusts in Germany

Susanna Mohr[1,2], Michael Kunz[1,2], Alexandra Richter[3], and Bodo Ruck[2,3]

[1]Institute of Meteorology and Climate Research (IMK-TRO), Karlsruhe Institute of Technology (KIT), Karlsruhe, Germany
[2]Center for Disaster Management and Risk Reduction Technology (CEDIM), Karlsruhe, Germany
[3]Institute for Hydromechanics, Karlsruhe Institute of Technology (KIT), Karlsruhe, Germany

*Correspondence to:* Susanna Mohr (mohr@kit.edu)

**Abstract.** Due to the small-scale and non-stationary nature of the convective wind gusts usually associated with thunderstorms, there is a considerable lack of knowledge regarding their characteristics and statistics. In an effort to remedy this situation, we investigated in this study a set of 110 climate stations of the German Weather Service between 1992 and 2014 to analyze the temporal and spatial distribution, intensity, and occurrence probability of convective gusts.

Similar to thunderstorm activity, the frequency of convective gusts decreases gradually from South to North Germany. No further spatial structures, such as a relation to orography or climate conditions, can be identified regarding their strength or likelihood. Rather, high wind speeds of above $30\,\mathrm{m\,s^{-1}}$ can be expected everywhere in Germany with almost similar occurrence probabilities. A comparison of the 20-year return values of convective gusts with those of turbulent gusts demonstrates that the latter have higher frequencies, especially in northern Germany. However, for higher return periods, this effect can be reversed at some stations.

The values of the convective gust factors are mainly in a range between 1 and 4 but can even reach values up to 10. Besides the dependency from the averaging time period of the mean wind, the values of the gust factors additionally depend on the event duration and the storm type, respectively.

**Keywords:** convective wind gusts, monthly distribution, spatial distribution, return periods, convective gust factor, Germany

## 1 Introduction

Convective wind gusts in terms of non-tornadic, straight-line winds cause significant damage to buildings and other structures in many parts of the world. Because they occur in Central Europe predominantly in the warm summer months, they may pose a significant threat to outdoor activities or open-air concerts. An example of such damage cases is the music festival *Pukkelpop* (Hasselt, Belgium) in the summer of 2011 with estimated wind speeds of between 29 and $37\,\mathrm{m\,s^{-1}}$, at which five people died and at least 140 were injured due to a stage collapse (De Meutter et al., 2015). Another recent example is the Pentecost storm of 9 June 2014 connected with an intense bow echo in Western Europe with high wind speeds up to $40\,\mathrm{m\,s^{-1}}$ that caused six fatalities and total losses of 650 million Euros (Barthlott et al., 2017; Mathias et al., 2017).

Various observations have shown that convectively induced storm events can reach even higher wind speeds compared to large-scale wind storms. Wind peaks of more than $50\,\mathrm{m\,s^{-1}}$ have already been recorded (Proctor, 1993; Lemon, 1998; Dotzek

et al., 2009). The highest convective gust ever observed was in 1983 during a microburst in the United States with a peak gust of $67\,\mathrm{m\,s^{-1}}$ (Fujita, 1990). Based on damage assessments, one can assume that wind speeds comparable to those of an F3 tornado are possible.

Severe convective gusts are caused either by local-scale downbursts that create strong divergent horizontal winds near the ground or by meso-scale cold pools associated with horizontal pressure gradients large enough to produce high winds speeds in the absence of strong downdrafts (Wakimoto, 2001; Markowski and Richardson, 2010). In general, the development of convective gusts is determined by diabatic processes, dynamic and thermodynamic perturbation pressures, and precipitation loading, the latter mainly as a trigger (Proctor, 1989; Wakimoto, 2001).

Due to the local-scale nature of severe convective storms, however, existing observation systems do not record convective gusts accurately, homogeneously, and uniquely. In Germany, approximately 500 climate stations currently measure wind speeds, which means that one station has to cover an area of $700\,\mathrm{km^2}$ on average, which is far from being adequate. This leads to an underestimation of local-scale events, such as microbursts that have only a small chance of being recorded, compared to larger events, such as gust fronts (Fujita, 1990; Orwig and Schroeder, 2007).

Thus, there is a considerable lack of knowledge regarding the characteristics, frequency, and climatology of convective gusts. Furthermore, their amplification by urban structures and their influence on buildings is not yet fully understood (Richter et al., 2016a, b). According to this, convective wind events are not included in the present wind load standards of buildings and structures, which so far have been based solely on the characteristics of synoptically driven wind gusts in the near-surface boundary layer (e. g., DIN EN 1991-1-4:2010-12; ASCE, 2010). However, convective and turbulent gusts differ considerably concerning vertical wind-speed profiles, the gust factor describing the turbulence (i. e., maximum to mean wind speed), or exceedance probability curves (Wood et al., 2001; Letchford et al., 2002; Holmes, 2002; Chen and Letchford, 2005).

Several studies have already shown that convective gust factors are usually larger than those of turbulent gusts (Bradbury et al., 1994; Chay et al., 2008; Holmes et al., 2008; Lombardo et al., 2014; Shu et al., 2015; Solari et al., 2015). Choi and Hidayat (2002), for example, found values between 2.2 and 6.5 for convective gusts in open terrain in Singapore. In the United States, Orwig and Schroeder (2007) calculated a factor of 1.5 triggered by a rear-flank downdraft of a supercell and a value of 3.8 during a derecho. Regarding return values, Friederichs et al. (2009) have quantified the probability of wind gusts to exceeding a certain warning level in Germany, but they did not differentiate between convective and turbulent gusts. However, as already shown by Gomes and Vickery (1978), Bradbury and Deaves (1994), and Holmes (2002), both gust forms have very different exceedance probability curves and thus have to be separated for proper extreme value analysis. The wind gust assessment of Kasperski (2002), for example, defines convective gusts when the maximum gust speed is well above the mean wind. However, the author stated that thunderstorms have almost no influence on the strong wind climate, which could be due to its insufficient definition.

The motivation behind this study is to identify various characteristics of convective gusts by means of the statistical analysis of wind measurements in Germany, for which only limited information is available at present. A question in this context is which temporal and spatial properties convective gusts exhibit and if their distributions are similar to other severe convective events. Besides the diagnosis of maximum wind speeds, we estimated the occurrence probabilities of extreme convective gusts

for larger return periods (e. g., 50 or 100 years) and discussed those in comparison to turbulent gusts. Finally, we systematically investigated convective gust factors based on a long-term data set of more than 23 years, which has not yet been considered.

The paper is structured as follows: Section 2 gives a short overview of the data sets and statistical methods used in this study. Furthermore, it describes the filtering method for identifying convective gusts among all wind measurements. Sections 3.1 and 3.2 present the seasonal and spatial variability of past convective gusts in Germany. Section 3.3 investigates the return values of convective gusts for higher return periods and compares the results with those of turbulent gusts, while Section 3.4 discusses the results of the convective gust factor. Finally, the last Section 4 summarizes the results and gives some conclusions.

## 2  Data and methods

Because convective gusts associated with thunderstorms occur predominantly in the summer half-year in Germany (and Central Europe; Wapler, 2013; Anderson and Klugmann, 2014), our examinations were also restricted to that time period. As a result, a large fraction of turbulent gusts were already excluded in the data set.

### 2.1  Observation data

The study used measurements of the German Weather Service (Deutscher Wetterdienst, DWD) from two operational networks: (a) climate data of the "KL network" with hourly measurements over a period of several decades, and (b) automatic measurements with a 10-minute resolution from the "MN network". The latter started only after 1990 and has been steadily expanded over the past two decades. Whereas only between 100 and 150 stations were in operation at the end of the last 20th century, the number quadrupled up to 2014.

The climate data (KL) were considered for the 23-year period between 1992 and 2014. We used both daily maximum wind gusts (FX) and hourly mean wind speeds (FF). For both data sets, the DWD performed a basic quality control. Only stations where observations were available for the complete 23-year period with a data loss below 10 % were evaluated. Furthermore, we excluded stations located in the North Sea or Baltic Sea or above 900 m.a.s.l., where high wind gusts are strongly influenced by orographically induced flow effects and where a separation between synoptically and convectively driven gusts is problematic (mixed wind climate). In total, we considered the remaining 110 stations, resulting in a station density of approximately one station per $3250\,km^2$. Even though this density is rather low, the selected stations are almost evenly distributed across the investigation area and well represent the terrain characteristics.

In addition to wind speed, daily pressure records (KL data) were used to filter out turbulent gusts (see Sect. 2.4). For this, the pressure differences among six climate stations located over Germany were considered (Schleswig, Norderney, Hannover, Berlin-Tempelhof, Frankfurt/Main-Airport, Hof, and Augsburg). In all case, the distance between two neighboring stations is less than 250 km, which ensures that also smaller-scale depressions are captured.

For a more detailed discussion of convective gust factors, we additionally used the maximum gust and mean wind every 10 minutes (MN data). Only stations and days were considered that exist in both data sets.

## 2.2 Return values of turbulent gust

For the comparison of the occurrence probability between convective and turbulent gusts (Sect. 3.3), we used the return values related to winter storms from the study of Kunz et al. (2010). The return values for return periods between 1 and 100 years were calculated from KL data during the winter half-year between 1971 and 2000 using the same statistical method as applied in this study (see Sect. 2.5). In total, the data of 85 out of the 110 stations were available. For more details, the reader is referred to Kunz et al. (2010).

## 2.3 Lightning data

Data from the lightning information service "BLitz InformationDienst Siemens" (BLIDS), which is part of the EUropean Cooperation for LIghtning Detection network (EUCLID; Schulz et al., 2016; Drüe et al., 2007), were used to ensure that wind measurements were related to thunderstorm events. We considered only the time and location of the lightning, whereas polarity and power information was neglected. Because cloud-to-cloud (CC) lightning was not recorded entirely due to the lower frequency range (Drüe et al., 2007), only cloud-to-ground (CG) lightning was taken into account for identifying thunderstorm days. Although three-quarters of all lightning is CC (Rakov and Uman, 2003), we assume that severe convective storms with strong downdrafts and thus strong gusts at the surface produce a sufficient number of CG flashes. Lightning detection has been available since 1992 in Germany, whereby the replacement of the lightning sensors in 2000 did not affect our filter approach.

## 2.4 Definition of convective gusts

The separation and classification of strong wind events into synoptically/frontal driven and convectively driven gusts is a key topic, because any inadequacies in this stage may severely affect subsequent statistics. Different approaches have been established that are either based on individual detailed inspections of the prevailing meteorological conditions (by using surface measurements, radar/satellite images,...) during the wind events or on systematic filtering using proxy data for thunderstorms such as lightning. A summary of the different methods to extract convective gusts can be found in De Gaetano et al. (2014).

We follow the approach of Choi and Hidayat (2002) and Cook et al. (2003) and considered wind gusts only in combination with a thunderstorm day. In this process, a thunderstorm day is classified when at least five lightning flashes occur in a box of $10 \times 10\,\mathrm{km}^2$ (Piper and Kunz, 2017). Such a thunderstorm day has to be within a 50-km radius around the wind record. The reason for the large detection radius is that gust fronts can occur several tens of kilometers ahead of a storm center with lightning activity, as already shown by Klingle et al. (1987), Houze (2014), and Pantillon et al. (2015). Note that a lower distance does not modify the overall results.

However, the occurrence of lightning (or thunder) does not necessarily mean that a severe wind gust was generated by a thunderstorm cloud. In fact, low pressure systems and related cold fronts, especially during the summer, may also produce severe gusts. Because the driver is a mixture of convectively driven processes and pressure gradient in those cases, we wanted to exclude such events from the sample. Therefore, our filtering additionally eliminates events that are related to large pressure gradients. In a first step, we excluded high-wind events on days with strong pressure gradients of $4\,\mathrm{Pa\,km}^{-1}$ or more in

proximity to the wind station. Means that pressure gradients between the nearest (to the gust event) pressure station and the others five pressure stations (cf., Section 2.1) are considered. Because storm events related to low pressure systems frequently occur in April on the north coast, we used for this month and north of 52°N (North German Plain) a filter that excludes days under the influence of weaker pressure gradients compared to the first case ($> 2.5 \, \mathrm{Pa \, km^{-1}}$). This criterion was extensively tested by a comparison with s synoptic weather charts.

Both the distance to a lightning recording and the thresholds of pressure gradients were identified by sensitivity and individual case studies by investigating the impact of varying thresholds on the sample size and the following results. In the case of considering only events greater than or equal to $18 \, \mathrm{m \, s^{-1}}$, the two filters together lead to a reduction in data of 63 %, whereas the pressure filter constitutes only a small fraction (1 %). This procedure reduces the sampling, but the resulting event set is still a representative basis for our assessment.

## 2.5 Extreme value statistics

Basically, two different methods exist for describing extreme events, i.e., for assessing the tail behavior of the distribution of interest. One is the classical generalized extreme value (GEV) distribution, which comprises three different distributions considering only annual maxima (Fisher and Tippett, 1928; Jenkinson, 1955). The other approach is the peaks-over-threshold (POT) method, at which all events above a chosen threshold $\zeta$ are selected for calculating the upper tail of a sample with the generalized Pareto distribution (GPD; Palutikof et al., 1999; Coles et al., 2001). This method increases the number of considered events and thus reduces statistical uncertainty (Brabson and Palutikof, 2000). After testing both methods, we chose the POT/GPD method to relate wind speeds and probability, as that approach reproduced the observed gusts better. Note, however, that the differences between the return values as considered in this work ($< 100$ years) estimated from both methods are considerably smaller than the uncertainties of the method itself (here expressed by confidence bounds).

The cumulative distribution function (CDF) of the GPD is

$$F(x) \quad = \quad 1 - [1 + \frac{k}{\sigma}(x - \zeta)]^{-1/k} \qquad \text{if } k \neq 0 \tag{1}$$

$$F(x) \quad = \quad 1 - \exp\left[-\frac{(x - \zeta)}{\sigma}\right] \qquad \text{if } k = 0 \quad , \tag{2}$$

where $x$ is the random variable (cf., Coles et al., 2001). The shape parameter $k$ indicates the width, and the scale parameter $\sigma$ indicates the slope of the CDF. Both parameters were estimated using the maximum likelihood method as a reliable and robust estimator (Hosking and Wallis, 1987; Kunz et al., 2010). The parameter $k$ is typically valid within the range of $-0.5 < k < +0.5$ (Abild et al., 1992).

With the crossing rate $\lambda$ as the expected number of peaks above $\zeta$, wind speed is a function of the return period $T$

$$X_T \quad = \quad \zeta + \frac{\sigma}{k}[(\lambda T)^k - 1] \qquad \text{if } k \neq 0 \tag{3}$$

$$X_T \quad = \quad \zeta + \sigma \ln(\lambda T) \qquad \text{if } k = 0 \quad . \tag{4}$$

For $k < 0$, the function converges asymptotically toward an upper bound. However, it is infinity if $k \geq 0$, implying an unbounded increase in wind speed for increasing return periods, which is physically unrealistic (Holmes and Moriarty, 1999).

The application of the POT/GPD requires that the event set be independent and identically distributed. Therefore, we used the daily wind maximum assuming that two events at midnight do not occur. This is in agreement with Lombardo et al. (2009), who found a negligible dependence on whether the minimum time distance between successive convective gust peaks was between 6 or 12 h.

5    To consider statistical uncertainty, the 95 % confidence bounds are calculated by bootstrapping. This method is based on a number $n$ of samples—here $n = 1000$ times—obtained by random resampling with the replacing of the original data set (Efron and Tibshirani, 1993).

Because $F(x)$, $k$, and $\sigma$ are strongly controlled by the choice of $\zeta$, this quantity must be adapted to the climatology of the respective data set (here, for each station). We applied two different techniques for the threshold selection (Coles et al., 2001):

10    First, we plotted the mean residual life graph, also known as conditional mean exceedance, which shows the mean excess over a threshold $E(x - \zeta \,|\, x > \zeta)$. The idea is to find the lowest threshold, at which the graph is nearly linear (constant slope)—taking into account the 95 % confidence bounds. Second, by using the parameter stability plots ($k$ or $\sigma$ against varying $\zeta$), a threshold can be identified, where both parameters above are almost constant. Note, however, that despite the thorough determination of $\zeta$, the results still depend (slightly) on that parameter, whereby the differences are small and within the confidence bounds.

## 15  3   Statistical characteristics of convective gusts

Focusing on severe wind events, we considered in the following particularly gusts greater than or equal to $18 \, \mathrm{m \, s^{-1}}$, which corresponds to the warning criterion level 2 of DWD (warning of significant weather). In Section 3.1, a second threshold of $25 \, \mathrm{m \, s^{-1}}$ was used, defining severe wind gusts in the United States according to the National Oceanic and Atmospheric Administration (NOAA).

### 20  3.1   Seasonal variability

Similar to other convective-related phenomena in Germany, such as lightning, hail, or tornadoes (Wapler, 2013; Piper and Kunz, 2017; Puskeiler et al., 2016; Groenemeijer and Kühne, 2014), convective gusts occur predominantly in the warm summer months between May and August, when atmospheric conditions favor the formation of deep moist convection. Most events of the sample exceeding a threshold of $18 \, \mathrm{m \, s^{-1}}$ (in total 5274 events; KL data) are observed in June and July (53 %) with the

25    absolute maximum in the second half of June and July (11-day running mean in Fig. 1a). Occurrence probability is similar in May and August (17/18 %), while it is in the order of 6 % in April and September. Note that we considered every single measurement at each station, which means that one event can be recorded on two or more stations if the event is related to a meso-scale cold pool.

The monthly distributions considering higher thresholds are very similar (Fig. 1a). Severe events ($\geq 25 \, \mathrm{m \, s^{-1}}$) are slightly

30    more frequent in July, wherever the differences in June are negligible. However, due to the limited sample size typically connected with extremes, we observed a very high day-to-day variability (95 % confidence intervals; dashed lines in Fig. 1a). Furthermore, severe gusts occur very rarely in April or September, when prevailing atmospheric conditions do not usually favor

severe thunderstorms. Note that the peak at the beginning of April for both thresholds is caused by one single event (4 April 2001). On that day, a cold front with a large number of embedded thunderstorms moved over Germany and produced several convective gusts with velocities up to $32\,\mathrm{m\,s^{-1}}$.

By splitting the observations into a north and south region comprising 55 stations for each sample ($\sim 51.4°\mathrm{N}$), it is found that the maxima are shifted about 15 days against each other with the earlier occurrence in the north (Fig. 1b). However, this difference is observed only when individual events were considered. For the number of convective wind gust days, this shift vanishes (not shown). Furthermore, it is striking that the differences between the north and south are pronounced in May, July, and August but not in June. Unfortunately, underlying causes have not yet been identified.

Interesting is the drop in the number of convective gusts around 15 June for both thresholds and in the two regions. One explanation for this drop could be the cold spell, which is frequently observed in the middle of June in Central Europe (in German: "Schafskälte"), and which may reduce thunderstorm activity.

Our results are roughly consistent with previous studies considering convective gusts in Germany: in particular, the growth after April due to increasing temperature and decreasing stability, the highest activity in the two months of June and July, and the subsequent decrease at the end of the summer. However, Tous and Romero (2006), Dotzek and Friedrich (2009), and Gatzen (2013) observed significantly higher gust activity in August than in May. Furthermore, all three studies disagreed about whether June or July exhibits the highest number of convective gusts. Differences in our study are the data basis (European Severe Weather Database, ESWD; Tous and Romero, 2006; Dotzek and Friedrich, 2009), the determination of the data basis (Gatzen, 2013), and the investigated time period.

## 3.2 Spatial distribution

In accordance with the thunderstorm activity (Piper and Kunz, 2017), convective gusts exceeding a lower threshold of $18\,\mathrm{m\,s^{-1}}$ occur more frequently in the southern half of Germany compared to the north (Fig. 2a). Whereas the overall number of those events at the northern stations is $40 \pm 17$ on average ($\pm$ standard deviation) during the 23-year period, the number is $56 \pm 22$ in the southern part.

This north-to-south gradient, however, cannot be observed in the analysis of high percentile values. The 95 % percentile values (Fig. 2b) are on average $18.9 \pm 1.3\,\mathrm{m\,s^{-1}}$ but show only slight variability over Germany. Furthermore, a relation between percentile values and orography, as is the case for turbulent gusts (Hofherr and Kunz, 2010; Kunz et al., 2010), cannot be established (Fig. 3). Moreover, local minima or maxima seem to be the result of the stochastic nature of convection. Note that the 95 % percentile represents approximately between 10 and 30 gusts per station in 23 years in North Germany and between 35 and 50 in South Germany.

Almost the same applies to the distribution of the peak values per station (Fig. 2c). Again, distinct spatial differences of maximum convective gust speeds or relationships to orography are not found. Figure 2c rather suggests that peak values exceeding $30\,\mathrm{m\,s^{-1}}$ can be expected everywhere in Germany (on average: $30.2 \pm 4.4\,\mathrm{m\,s^{-1}}$). The strongest convective gust in the past two to three decades was $52.6\,\mathrm{m\,s^{-1}}$ (Fig. 2c blue dot) and was recorded on 29 July 2005 in Zinnwald-Georgenfeld (Saxon Switzerland-East Ore Mountains).

### 3.3 Return values and periods

Estimating with the POT/GPD method the occurrence probability of extreme wind gusts for return periods larger than the investigation period, the choice of the threshold $\zeta$ must be performed carefully. Because each station has its own climatological background and underlying distribution function, the shape and scale parameters, $k$ and $\sigma$, can show clear differences among all locations. Because the sample sizes of the stations are different, the use of percentile values appears to be more reasonable than implementing a fixed number of the strongest events.

Based on the mean residual life graph and parameter stability plots (both for $k$ and $\sigma$) for each station, we identified the 80 % percentile as an appropriate threshold $\zeta$, resulting in values between 12.5 and 17.0 m s$^{-1}$ for the majority of stations (Fig. 4d). Only two stations exhibit thresholds below this range. As presented in Figures 4a–c, this range fulfills the criteria of both techniques—conditional mean exceedance and parameter stability plots—very well (see Sect. 2.5). For example, for the station of Munich-City (blue dot in Fig. 4b and c), the $k$ and $\sigma$ stability plots show almost constant behavior for slightly varying threshold values above the resulting value of $\zeta = 14.6$ m s$^{-1}$ (red lines in Fig. 4b and c).

Based on all selected 110 stations in Germany, the return values of convective gusts for return periods of 20 years (RV$_{20a}$) and 50 years (RV$_{50a}$) are on average $27.8 \pm 2.5$ m s$^{-1}$ and $30.2 \pm 3.1$ m s$^{-1}$, respectively (Fig. 5). In addition, isolated high values are estimated between 32 and 36 m s$^{-1}$ for RV$_{20a}$ and between 36 and 40 m s$^{-1}$ for RV$_{50a}$. High return values of up to 50 m s$^{-1}$ for RV$_{50a}$ as estimated by Lombardo (2012) for the West Texas region (United States) cannot be estimated for Germany. Note, however, that a value above 50 m s$^{-1}$ has already been observed (cf. Fig. 2c, Zinnwald-Georgenfeld), but with an extremely high estimated return period of approximately 1,000 years (see also Seregina et al., 2014, supplemental material Table 1). Depending on the respective probability density function (or the parameters $k$ and $\sigma$; see Fig. 4e and f), the increase between both return values substantially differs among the stations. Although the differences between RV$_{20a}$ and RV$_{50a}$ are on average $2.3 \pm 0.9$ m s$^{-1}$, there are individual stations with differences between 4 and 6 m s$^{-1}$ (e.g., Lahr; blue dot in the southwest corner of Fig. 5b). Furthermore, a comparison with Figure 2b shows that stations with similar 95 % percentile values do not have to show similar return values influenced by the underlying probability density function as well. For example, convective gusts with a one year return period (RV$_{1a}$) correspond—depending on the station—to its 95 % to 98 % percentiles (correlation of 0.97).

Depending on the station and thus on the individual shape parameter $k$, the uncertainty due to the application of an extreme value statistic is in a range between 3 and 16 m s$^{-1}$ with a median of 6.5 m s$^{-1}$ for RV$_{20a}$ and between 4 and 25 m s$^{-1}$ with a median of 9.5 m s$^{-1}$ for RV$_{50a}$ (not shown). Higher values of confidence bounds are primarily caused by positive $k$ values (semi-logarithmic connection, see Eq. 4), leading to an unbounded behavior of the upper distribution function. This is due to the fact that $k$ depends on the range of the data sample considered for the GPD calculation. Especially single maxima or individual outliers result in positive $k$ values (not shown). As shown in Figure 4e, the maximum likelihood method estimates for 32 % of all stations positive $k$ values.

Figure 6 summarizes the results for five return periods (1, 10, 20, 50, and 100 years) in four regions of Germany. As expected, statistical uncertainty (black bars) increases with return periods, mainly caused by stations with positive $k$ values. Furthermore,

the confidence bounds are in most cases—except for one year events—higher than the regional variability (red lines). Due to the limited data availability of 23 years, the results for the 100-year events exhibit high uncertainties (on average $\sim 14\,\text{m}\,\text{s}^{-1}$) and should be handled with care. In summary, convective gusts above 20 (25) $\text{m}\,\text{s}^{-1}$ are on average observed throughout Germany each (10) year(s). This result is similar to a study of Holmes (2002), who showed that the occurrence rate of downbursts

exceeding $21\,\text{m}\,\text{s}^{-1}$ is around one event per year at stations in the Melbourne area in Australia.

Both Figures 5 and 6 show a tendency of slightly higher mean return values for southern stations compared to those in the north. For example, $\text{RV}_{20a}$ is on average $27.0 \pm 2.0\,\text{m}\,\text{s}^{-1}$ in the north and $28.7 \pm 2.6\,\text{m}\,\text{s}^{-1}$ in the south (Fig. 5a). However, the differences are smaller than the variability of the stations in the respective areas or statistical uncertainty due to the application of the statistical method, respectively. In agreement with the percentile values or the maxima (Fig. 2 and Fig. 3), no specific

pattern related to the terrain or climate conditions can be identified. The convective gust intensity depends primarily on characteristics detached from the surface, as the intensification forces are particularly the mid-tropospheric flow, vertical temperature gradient, cloud water content, droplet size, and rain fall velocity (Proctor, 1989; Wakimoto, 2001). Surface roughness, which is essential for turbulent gusts, seems to be less important for convective gusts (Solari et al., 2015).

Comparing our results with return values for turbulent gusts as estimated by Kunz et al. (2010) and Hofherr and Kunz (2010)

for winter storms in Germany, it is found that the $\text{RV}_{20a}$ of the turbulent gusts are significantly larger (average: $7.3 \pm 3.9\,\text{m}\,\text{s}^{-1}$; Fig. 7). The difference in the north ($9.0 \pm 3.2$) is more pronounced than that in the south ($5.6 \pm 3.8$). This difference is mainly due to the fact that the return values of turbulent gusts show a distinct north-to-south gradient over Germany caused by the higher frequency of low pressure systems coming from the Atlantic Ocean in the north (Ulbrich et al., 2009; Feser et al., 2015). Thunderstorm activity in that area, on the other hand, is substantially reduced (Piper and Kunz, 2017) due to the higher stability

near the North Sea and Baltic Sea (Mohr and Kunz, 2013).

Finally, in some cases, the return values of convective gusts may become larger than those of turbulent gusts for higher return periods. This effect strongly depends on the underlying distribution function (not shown). For example, for $\text{RV}_{100a}$, this applies for 13 out of the 85 stations (15 %) considered for this comparison, which all (except one) are situated south of 52°N. These findings point out the need to distinguish between the two types of gusts for their statistics as already mentioned by Gomes and

Vickery (1978).

### 3.4   Thunderstorm or convective gust factor

Especially in the field of wind engineering for calculating structural design or wind loadings (Davenport, 1967), gust factors are commonly used to estimate the relation between gusts and mean wind. This relation for turbulent gusts depends on various effects, such as the surface roughness of the surroundings, thermal stability, or wind shear, mainly affecting the turbulence

characteristics of flows. In the case of convective gusts, however, gust factors, which will be estimated in the following based on the comprehensive available data set of the 110 stations in Germany, are severely affected by the averaging period of the mean wind (Lombardo et al., 2014).

The gust factor $\text{GF}_t$ considered here represents the ratio of the maximum wind speed $v_{max}$ to the mean wind speed $\overline{v_t}$ over an averaging time period $t$ (Holmes, 2015; Solari et al., 2015). Usually, $v_{max}$ is averaged over 3 s in accordance with the World

Meteorological Organization (WMO, 2010):

$$\text{GF}_t = \frac{v_{max}}{\overline{v_t}} \quad , \tag{5}$$

with time periods $t = 10$-min or 1-hr to calculate $\overline{v_t}$ and $\text{GF}_t$. Here, we used for $\overline{v_t}$ both MN (10-min) and KL data (1-hr). To enable a direct comparison of GF for the two averaging periods, we considered only events for which both data (MN and KL) are available. In total, 4,658 convective gusts with $\geq 18\,\text{m s}^{-1}$ were selected, corresponding to 88 % of cases in the KL data.

The convective gust factors, $\text{GF}_{10min}$ and $\text{GF}_{1hr}$, are substantially larger than typical values of the (turbulent) gust factors (Fig. 8). Depending on the land use, surface roughness, or atmospheric stability, respectively, turbulent gust factors fluctuate usually between 1.2 and 2.3 (Wieringa, 1986; Brasseur, 2001; Hofherr and Kunz, 2010). In particular, $\text{GF}_{1hr}$ with an average of $2.9 \pm 1.0$ is well above that range (Fig. 8a blue). For $\text{GF}_{1hr}$, single values between 6 and 10 are even computed (1 %). In contrast, $\text{GF}_{10min}$ shows lower values with an average of $2.1 \pm 0.8$ (Fig. 8a pink). A simple fit (non-parametric) demonstrates that the spread of $\text{GF}_{1hr}$ is nearly twice that of $\text{GF}_{10min}$. The reason for the considerable differences between the turbulent and convective gust factors is that the latter strongly depends on the event duration or on the storm type substantially affecting the mean wind.

Comparable results and values have already been observed in other studies, which, however, were not based on such a large sample as that considered in this work (e. g. Orwig and Schroeder, 2007; Holmes et al., 2008; Shu et al., 2015; Solari et al., 2015). Choi and Hidayat (2002), for example, found $\text{GF}_{1hr}$ values between 2.2 and 6.5 for convective winds in open terrain in Singapore. They also stated that gust factors obtained close to the storm center may reach values between 7 and 8. Lombardo et al. (2014) estimated with an empiric function values between 1.5 and 3 for an average period of 10-min and between 3 and 6 for hourly data. Based on a sample of approximately 550 events, Durañona (2015) calculated $\text{GF}_{10min}$ values between 1.2 and 4.1.

Comparing $\text{GF}_{10min}$ and $\text{GF}_{1hr}$ for all single events shows significant differences between both (Fig. 8b). In 90 % of all cases, $\text{GF}_{1hr}$ is greater than $\text{GF}_{10min}$ (on average $1.0 \pm 0.8$). It is interesting to note that $\text{GF}_{10min}$ values in 10 % of the cases are larger than $\text{GF}_{1hr}$ values. The reasons for this counterintuitive behavior are the fixed defined measurement intervals and the event duration. If a gust occurs at the end of a 10-min interval, lower wind measurement at the beginning can lead to lower 10-min mean wind speeds compared to a 1-hr mean. A calculation by a running-average method prevents this effect (Chay et al., 2008; Solari et al., 2015; Mohr et al., in prep.). This is, however, only possible when high resolved measurements are operationally archived, which is currently not the case.

## 4 Discussion and conclusions

This study has examined the statistical characteristics of convective gusts in terms of non-tornadic, straight-line winds in Germany with a focus on their temporal and spatial distribution, including their intensities and occurrence probabilities. The data basis was the wind measurements of 110 climate stations of the German Weather Service (DWD) between 1992 and 2014 in the summer half-year. First, we generated a convective gust event set by filtering all wind measurements with lightning

and daily pressure gradient data to ensure the conjunction with downbursts, gust fronts, or cold pools of thunderstorm events and to exclude a connection to low pressure systems and related cold fronts. Afterward, we investigated monthly and spatial distributions of past events exceeding a lower threshold of $18\,\mathrm{m\,s^{-1}}$, which corresponds approximately to the 95 % percentile of the filtered wind data set. By applying peaks over threshold (POT) in conjunction with the generalized Pareto distribution (GPD), we estimated the probabilities of extreme events for different return periods and compared the results with those for turbulent gusts. Finally, convective gust factors (based on both 10-min and hourly mean wind speeds) were calculated based on a long-term event set.

The following major points and conclusions can be inferred from the results obtained:

1. Most of the convective wind gusts occurred during the warmest months June and July in accordance with other phenomena associated with thunderstorms. However, the monthly distributions of northern and southern stations show some discrepancies, such as the occurrence of maxima.

2. The frequency of convective gusts is higher in South Germany compared to the north. However, the intensity or probability distribution in terms of percentiles, maximum values, or return values show neither a large-scale gradient related to the climate conditions nor local-scale variability related to orography. Quite the reverse: High wind speeds above $30\,\mathrm{m\,s^{-1}}$ can be expected everywhere in Germany with a similar occurrence probability.

3. A comparison of the 20-year return values of convective gusts with those of turbulent gusts demonstrates that the latter have higher return values. This might be explained by the fact that more stations are affected during winter storms compared to thunderstorm events, resulting in an increased representativity. For higher return periods, this effect, however, can be reversed at some stations. Furthermore, the difference between both gust types is more pronounced in the north, where deep depressions coming from the Atlantic play a more major role than thunderstorm activity does.

4. Quantified from the comprehensive convective gust event set, the convective gust factors (GF) show on average values of $2.1 \pm 0.8$ (10-min mean wind) or $2.9 \pm 1.0$ (hourly mean wind), which are well above the range of turbulent gust factors. Furthermore, single values between 6 and 10 were even observed. The broad range demonstrates the large temporal variability of convective storms. Besides the dependency of the mean wind by the averaging time period, the values strongly depend on the event duration or storm type. As already noted by Durañona (2015), the convective gust factor may be a good additional indicator to separate strong wind gust caused by severe thunderstorms from a mixed wind climate.

A potential weakness of our paper is the representation of the comparatively low number of stations. This weakness cannot, however, be overcome, but growth would continue in the future, when the ongoing reduction of observation stations is maintained.

Important to keep in mind is the fact that, due to thunderstorms characteristics, current monitoring systems tend to underestimate the intensity (and also the likelihood) of convective events (see also Trapp et al., 2006). This leads to an underestimation

of the hazard and damage potential of those events, which may have important consequences for the wind load standards of buildings and structures. These have to be be revised accordingly by considering the characteristics of convective gusts.

Concerning extreme value statistics, the use of POT/GPD method is discussed controversially in the literature. Whereas the use is widely supported by some authors (e. g., Hosking and Wallis, 1987; Simiu and Heckert, 1996; Palutikof et al., 1999;
Holmes and Moriarty, 1999; Simiu, 2007), Harris (2005) demonstrated that this method has grave defects and thus is unreliable. Therefore, over recent years new methods have been developed to overcome this problematic, which are based, for instance, on long-term Monte Carlo simulations (Torrielli et al., 2013) or on the Penultimate distribution (Cook and Harris, 2004, 2008). These approaches for several cases may help to improve the statistics and to reduce underlying uncertainty.

## 5 Data availability

The hourly climate data (KL network) are freely available from the Climate Data Center (CDC) of the German Weather Service (DWD) via ftp (ftp://ftp-cdc.dwd.de/pub/CDC). The 10-minute data (MN network) are not freely available, but can be requested from the DWD. The lightning data can be requested from the lightning information service BLitz InformationDienst Siemens (BLIDS; http://blids.de).

*Acknowledgements.* The financial support of the German Research Foundation (Deutsche Forschungsgemeinschaft, DFG) for the presented
study (grant no. KU 1923/2-1 and Ru 345/35-1) is gratefully acknowledged by the authors. The authors thank the German Weather Service (DWD) for providing the surface measurements (regarding the MN data special thanks to Heiko Schmack), and the lightning information service "BLitz InformationDienst Siemens" (BLIDS) for providing the lightning data. We acknowledge the constructive and useful comments from Giovanni Solari and another anonymous reviewer that helped to improve the quality of our paper.

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

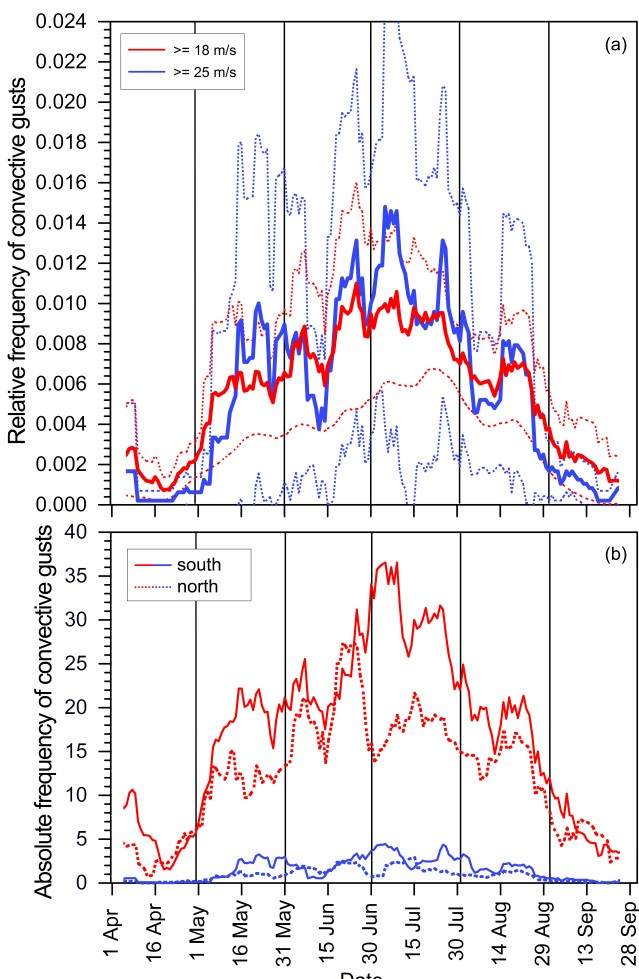

**Figure 1.** Mean seasonal distribution (running 11-day) of (a) relative (incl. 95 % confidence intervals; dashed) and (b) absolute frequency of convective gusts exceeding a threshold of $18\,\mathrm{m\,s^{-1}}$ (red) and $25\,\mathrm{m\,s^{-1}}$ (blue) considering (a) all stations in Germany, (b) stations north (dashed) and south (solid) of $51°\mathrm{N}$ (1992 – 2014).

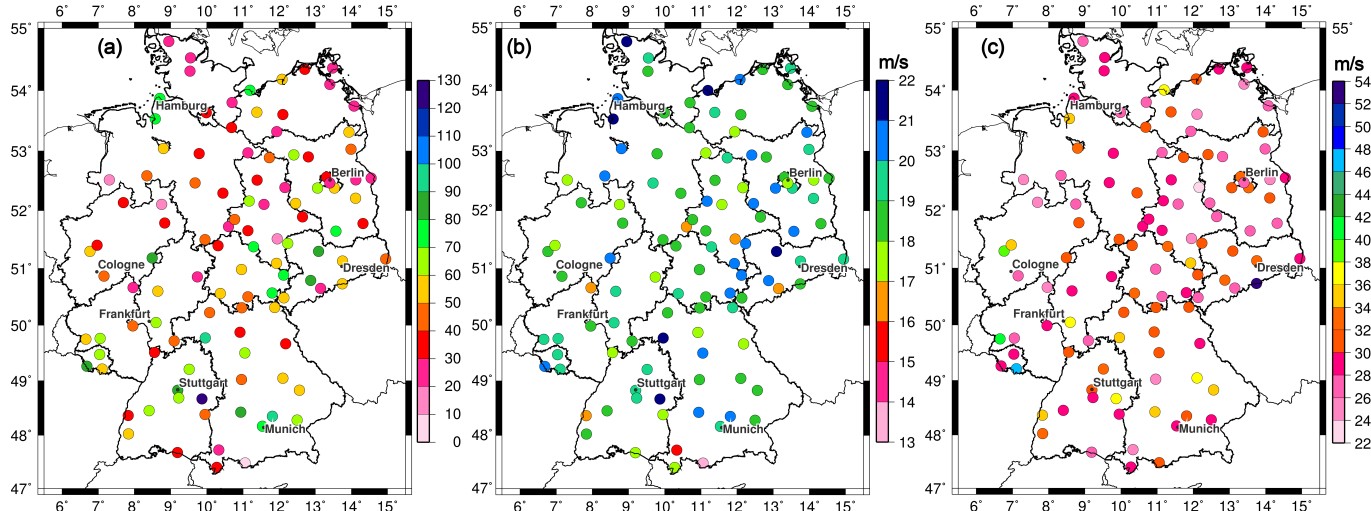

**Figure 2.** Spatial distribution of (a) the number of convective gusts exceeding a threshold of $18\,\mathrm{m\,s^{-1}}$, (b) the $95\,\%$ percentile of the convective gust sample, and (c) the maximum convective gust (1992 – 2014).

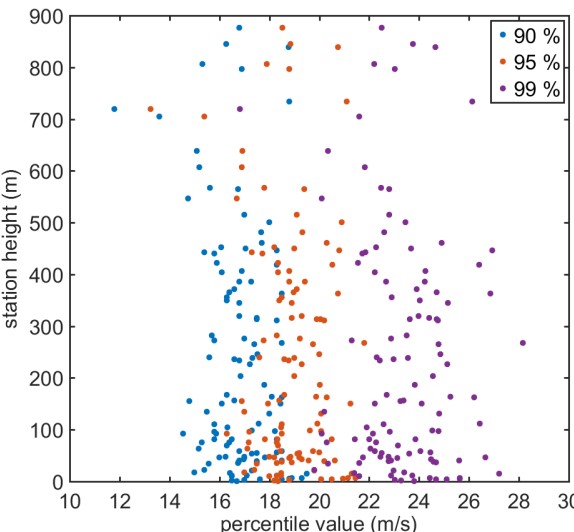

**Figure 3.** Scatterplot between different percentile values ($90\,\%$ in blue, $95\,\%$ in red, $99\,\%$ in purple) of the convective gust sample and the respective station height.

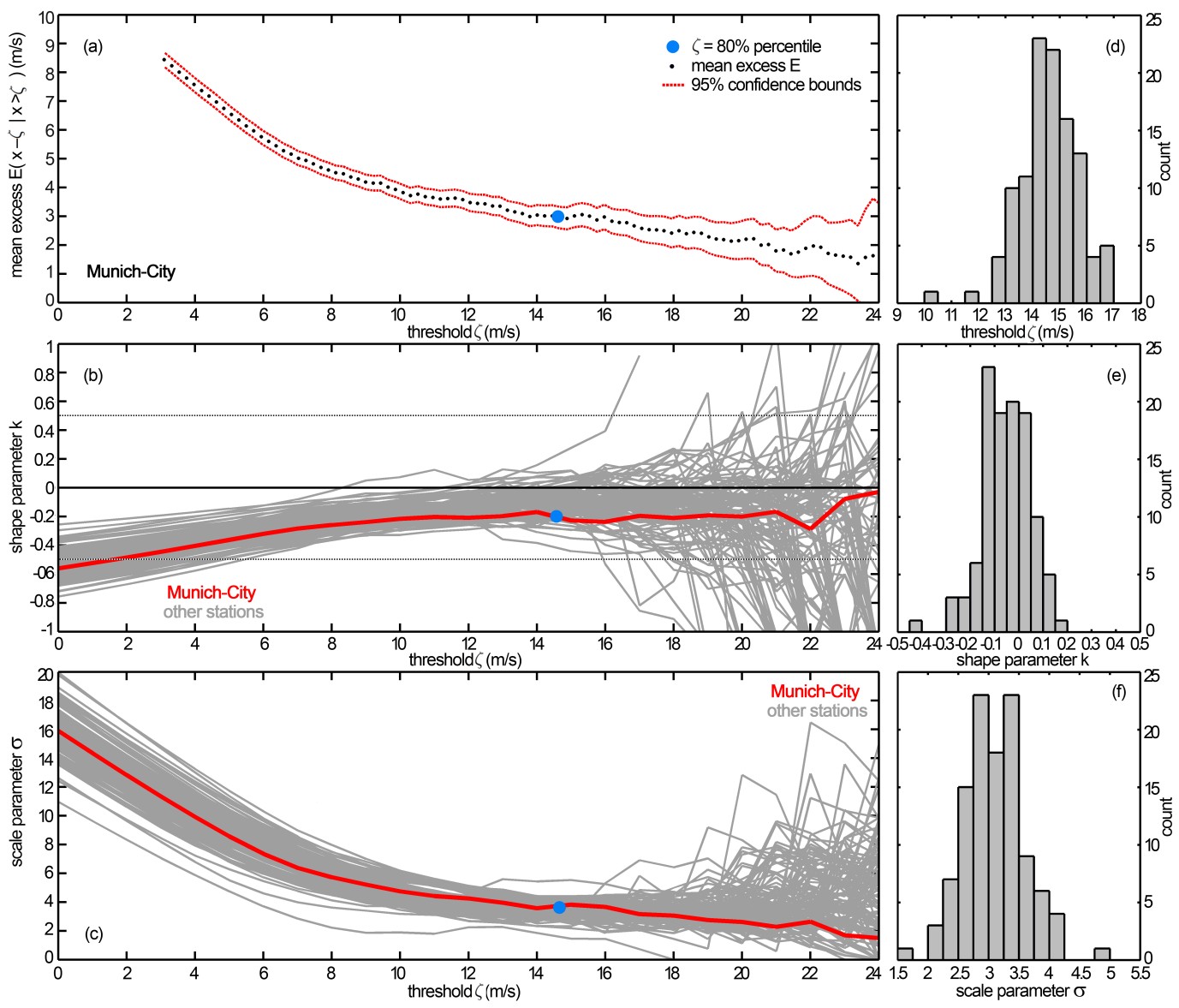

**Figure 4.** (a) Mean residual life plot exemplary for the station Munich-City with the 80 % percentile threshold (blue dot). (b) Stability plot of the shape parameter $k$ for all 110 stations (gray) and exemplary for Munich-City (red), and (c) as (b) for the scale parameter $\sigma$. Depending on the 80 % percentile of all 110 stations, the resulting histogram of (d) threshold $\zeta$, (e) shape parameter $k$, and (f) scale parameter $\sigma$ for the GPD.

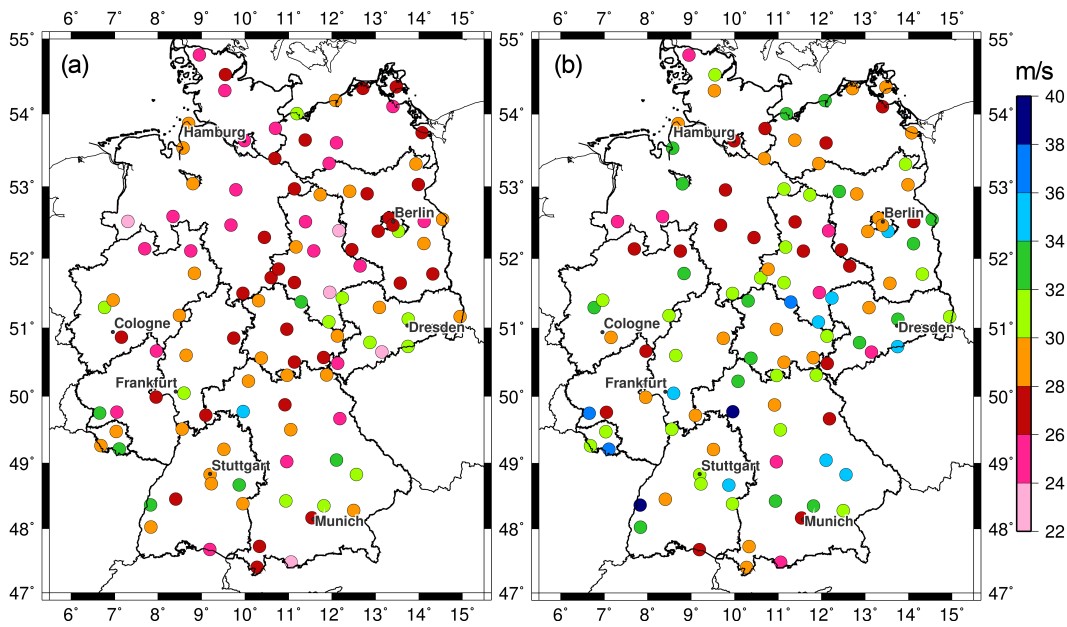

**Figure 5.** Spatial distribution of a convective gust for a return period of (a) 20 years ($RV_{20a}$) and 50 year ($RV_{50a}$).

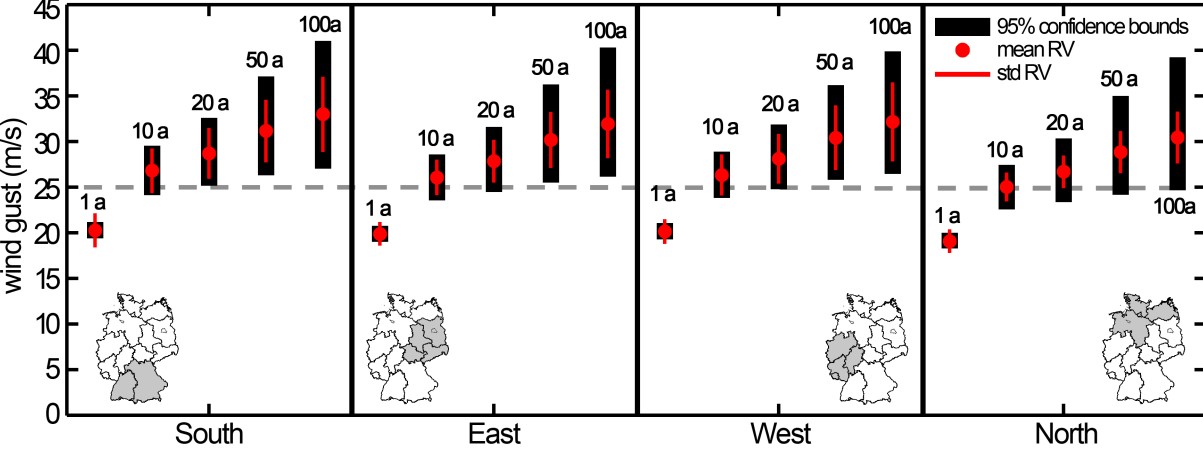

**Figure 6.** Mean return values of convective gusts (RV) for various return periods in four regions in Germany (gray area). Red lines indicate the standard deviation from all stations within the respective region, and black bars indicate the mean 95 % confidence intervals representing statistical uncertainties of the POT/GPD method.

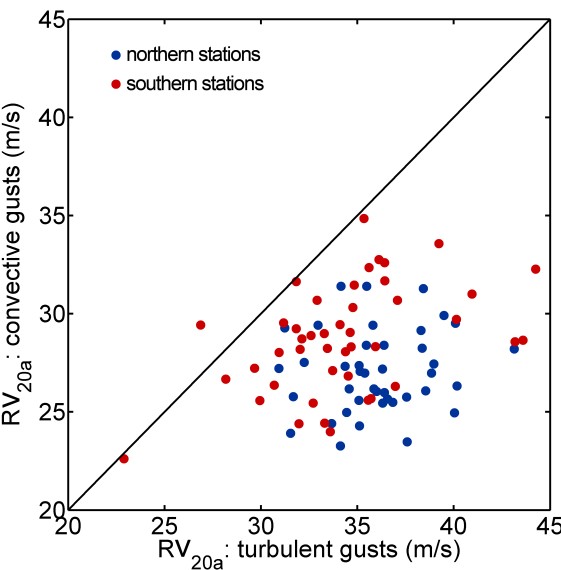

**Figure 7.** Scatterplot between the return value of a convective and turbulent gust for a return period of 20 years split into northern (blue) and southern stations (red).

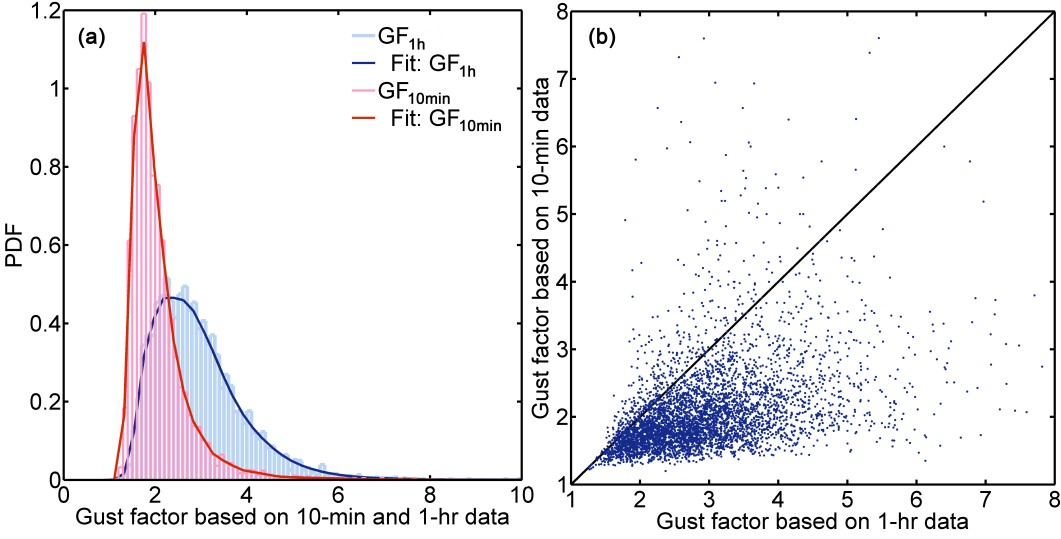

**Figure 8.** (a) Probability density function of convective gust factors based on both 1-hr measurements ($GF_{1hr}$; KL data) and on 10-min measurements ($GF_{10min}$; MN data), including a non-parametric fit. (b) Scatterplot between GF based on both MN and KL data.