# Peer review of "Statistical characteristics of convective wind gusts in Germany"

_Natural Hazards and Earth System Sciences, 2016_

## Referee Comment (RC1) · G. R. Solari (Referee) · 20 Feb 2017

Separating the occurrences and measurements of different Aeolian phenomena such as synoptic cyclones, thunderstorms, tornadoes and so on is a key topic of modern wind engineering in order to perform distinct statistical analysis, to extract the main statistical parameters related to each phenomenon, and to build wind field models suitable to represent the wind loading and response of structures. Merging these separate evaluations in a unitary formulation is a further aim still in the embryonic stage.

This paper provides very interesting and new information on several aspects in the above framework, thus represents a useful and pertinent contribution to the advance of the knowledge in this field. In its whole I appreciate it and support its publication.

This paper contains a broad literature both in the fields of atmospheric sciences and

wind engineering, perhaps a little biased towards the first field. Despite this I believe that some relevant contributions to this topic are not considered and some choices inherent the methods herein applied seem to be based on a limited view of some previous contributions. Under this point of view, without changing anything in the substance of this paper, I believe that a wider critical discussion on the advantages and shortcomings related to such choices may improve the quality of this paper and inspire future step forwards.

More in detail, I recommend Authors to take into account the following remarks and observations: • Section 1: at least two additional references should be considered. The first 1 is the fundamental paper that in 1977 introduced the concept of mixed wind climate and the idea of processing separately the statistical analysis of different wind phenomena. The second 2, published in 2002, deals with the same topic of the present paper just with reference to Germany. A comparison with previous methods and results is recommended. • Section 2: I am quite doubtful on the decision of restricting analyses to the summer half-year. In my experience thunderstorm events are concentrated in this part of the year but are present also, in minor proportion, all over the year. Restricting analyses to a period is even more dangerous considering the aim of performing a statistical analysis of the extreme wind speed. Unavoidably this produces underestimated results. I suggest to revise this choice in next contributions. • Section 2.1: Authors base their analyses on the daily peak and subsequent mean wind speeds on 10-min and 1-h periods. They also use pressure measurements. A very similar approach is used in Uruguay and described in 3. I suggest to examine this contribution. • Section 2.1: Also in the light of the occurrence of gust factors in the order of 6-10, I suggest Authors to consider the possibility that some peak values in the database may be wrong 4. The potential presence of some mistakes and the difficulty of recognizing them is a major shortcoming of this kind of analyses, where the control is very good in terms on mean values but almost impossible with reference to single peaks. • Section 2.3: I understand that Authors have probably no other opportunity than this use of lightning data. In my experience the presence of cloud-to-

cloud lightning not detected by measurements may provide some relevant drawback. I verified this by comparing similar lightning data with high-sampling velocity records. • Section 2.4: The problem of the separation of different wind events is a key topic because any mistake in this stage may compromise the quality of further evaluations. I suggest Authors to dedicate a few more words to this problem for instance using a citation to Lombardo et al. (2009) (included in references but not cited here) and to 5. • Section 2.5: Authors speak of GEV and POT/GPD and make the choice of using POT/GPD. This is fine but again, without changing the substance of this paper, this topic is a "world" that may necessitate a some more "delicate" approach. First of all the use of POT/GPD is widely supported by some Authors but drastically opposed by others. Ref. 5, for instance, is fully devoted to demonstrate that this method is wrong or at least unreliable. Our research group recently published a series of papers based on long-term Monte Carlo simulations 6,7 that confirms the limited reliability of the POT/GPD technique and arrives to the conclusion that the Process Analysis 8 (probably not easy to apply to thunderstorms) and the Penultimate distribution 9,10 are the best methods. • Section 2.5: At the end of this section Authors write "that the differences between the return values estimated by both methods are considerably smaller than the uncertainties of the method itself". This is absolutely correct with reference to return periods in the order of the number of years of available data, for instance 20-50 years. Structural safety, however, needs evaluations extrapolated to return periods in the order of 500-1000 years. Here, different methods lead to divergent results 9,10,11. • Section 3.1: Authors write: "we considered every single measurement at each station, which means that one event can be recorded on two or more stations". I think that this sentence may result misleading. Downburst are phenomenon with a radius of a few km. It is almost impossible that the same downburst may be detected by two stations of this network. The situation is different if Authors refer to the large scale wind event that generates downbursts. This point should be clarified. • Section 3.3: The last sentence deserves a citation to Authors that first expressed this concept 1. • Section 3.4: Line 11. The dependence

of the gust factor on the averaging period is discussed also by Solari et al. (2015).  c Section 3.4: Line 20. I do not agree on the sentence according to which "turbulent factors (to replace with gust factors ?) fluctuate usually between 1.2 and 2.3". Such a large variability necessarily depends not only on the roughness length but even more on the stability conditions. If wind is intense and of synoptic type, then atmosphere is neutrally stratified and the gust factor may vary between 1.25 and 1.75 with an average value around 1.5. In my opinion gust factors in the order of 1.75-2.3 may be ascribed to unstable conditions and intermediate events between large scale depressions and mesoscale downbursts 5.

Independently of the above remarks, that I wrote in a fully constructive spirit, I confirm my appreciation towards this contribution and that I consider it appropriate for publication. I hope that Authors may consider or discuss my remarks.

I suggest that this paper is accepted subjevt to minor revisions but I believe that a quick re-review may be useful.

Giovanni Solari Department of Civil, Chemical and Environmental Engineering Polytechnic School, University of Genoa, Italy

References 1 Gomes L, Vickery BJ (1977/1978). Extreme wind speeds in mixed climates. J Ind Aerod, 2, 331-344. 2 Kasperski M (2002). A new wind zone map of Germany. J Wind Eng Ind Aerod, 90, 1271-1287. 3 Duranona V (2015). The significance of non-synoptic winds in the extreme wind climate of Uruguay. Proc 14th Int Conf on Wind Engineering, Porto Alegre, Brasil. 4 Cook NJ (2014) Review of errors in archived wind data. Weather 69: 72-78. 5 De Gaetano P, Repetto MP, Repetto T, Solari G (2013). Separation and classification of extreme wind events from anemometric data. J Wind Eng Ind Aerod, 126, 132-143. 5 Harris, R.I., 2005. Generalized Pareto methods for wind extremes. Useful tool or mathematical mirage? JWEIA 93, 897-918. 6 Torrielli A, Repetto MP, Solari G (2013). Extreme wind speeds from long-term synthetic records, J Wind Eng Ind Aerod, 115, 22-38. 7

Torrielli A, Repetto MP, Solari G (2014). A refined analysis and simulation of the wind speed macro-meteorological components. J Wind Eng Ind Aerod, 132, 54-65. 8 Gomes, L., Vickery, B.J., 1977. On the prediction of extreme wind speeds from the parent distribution. J. Ind. Aerodyn. 2, 21-36. 9 Cook, J., Harris, I., 2004. Exact and general FT1 penultimate distributions of extreme wind speeds drawn from tail-equivalent Weibull parents. Struct. Saf. 26, 391-420. 10 Cook, J., Harris, I., 2008. Postscript to "Exact and general FT1 penultimate distributions of extreme wind speeds drawn from tail-equivalent Weibull parents". Struct. Saf 30, 1-10. 11 Lagomarsino, S., Piccardo, G., Solari G., 1992. Statistical analysis of high return period wind speeds, JWEIA 41, 485-496.

---

## Referee Comment (RC2) · Anonymous Referee #2 · 1 Mar 2017

General comments

The manuscript "Statistical characteristics of convective wind gusts in Germany" written by Susanna Mohr et al. describes a methodology to identify and select convective wind gusts from station measurements at 110 stations within Germany. Characteristics regarding the seasonality as well as spatial variations over Germany are considered and rare convective gusts are characterised by means of extreme value statistics. Additionally, by comparing the convective gust measurements to mean winds, gust factors are quantified.

Generally, the study presents very relevant work and is an important contribution to the understanding of local small scale convective wind gusts. The manuscript is well written and the chosen methods to assess the statistical characteristics of convective gusts in

general seem appropriate and well suited. However, I notice several minor flaws (which I listed below) in the methodological and statistical approach which I would recommend the authors to consider. I thus suggest the paper to be accepted after minor revisions.

Specific comments

P. 3, L. 17: Results show, that no significant differences are found in the intensity of rare convective gusts with respect to orography. Why are stations at higher ground excluded? It might be particularly worthwhile to also consider stations at higher altitudes!

P. 4, L. 13-14: The choice of a 50-km radius does not seem to be justified by the given explanation. Since a gust front can occur several kilometers ahead of a storm center this might suggest a radius of 5, possibly 10 km.

P. 4, L. 20 "proximity to the wind station": Pressure gradients are calculated by means of a small set of 6 climate stations. It should be explained how the pressure gradients "in proximity to the wind station" are determined and in how far it can be expected that small scale depressions can be captured (or why such small scale depressions are disregarded!).

P. 4, L. 20-22: This additional filter criterion seems a bit random/unsystematic. I suspect, that not only in April but also in autumn such weaker pressure gradients do occur. I would thus favor a more systematic treatment of seasonality. Also, this additional criterion might hinder the interpretation of spatial as well as the seasonal variance discussed later in the text.

P. 4, L 23-24: Sensitivity of what? It should be specified in which respect the sensitivity has been considered!

P. 5, L. 2: What is meant by "as the approach reproduced the sample better"?

P. 5, L 3-4: Does "uncertainties of the method itself" refer to confidence intervals on estimated return values? Should be clarified!

[Figure]

P. 6, L. 19-26: In Figure 1, I would propose adding confidence intervals to indicate the uncertainties in the seasonal variations. As noted above, a single event can cause the peak in beginning of April. Without a proper estimation of uncertainties (confidence intervals) I would challenge the statistical robustness of the results presented here!

P. 7, L. 7-9: Has this been tested explicitely or is this just an interpretation of the missing north-to-south gradient? Of course this could be explicitly done correlating orographic height of the station against percentile value?! This is also related to my previous comment on excluding stations at higher locations.

P. 7, L. 27-28: Although I do not want to object to the threshold choice itself, I do want to mention that this is not how the parameter stability criterion should be interpreted! In the GPD framework, it can be inferred that if the distribution of values above a certain threshold ($u\_0$) follows a GPD, then it follows a GPD above all thresholds higher than $u\_0$ with a modified sigma. Shape and modified scale should thus be constant above (not near) the chosen threshold within confidence intervals! For details see Coles: An Introduction to Statistical Modeling of Extremes, 2001 p. 78/79.

P. 8, L. 3-5: A comparison of the empirical estimates (95% percentile) and estimates from extreme value statistics might be interesting here. According to the numbers that are specified in lines 10-11 on page 7 we are then talking about a return period of about 1 year.

P. 8, L. 14-15: It should be clarified how the statistical uncertainty is calculated for a region. In the caption of figure 5 it is mentioned that it corresponds to the mean of 95% confidence levels. I do not see why and how this should compare to the standard deviation for different stations (regional variability)!

P. 10, L. 24: As mentioned in my previous comment, it should be clarified if this has been explicitly tested or weather this is simply the interpretation of Figures 2 and 4 (which do not contain an explicit information on orographic height).

P. 10, L. 26: By definition, an event with a 20-year return period has a fixed occurrence frequency of 20 years! Please rewrite!

Technical corrections

P. 7, L. 7: slight variability instead of slightly variability

––––––––––––––––––––

---

## Author Comment (AC1) · 27 Apr 2017

Dear Referees,

thank you very much for your work and the useful and valuable comments how to improve the scientific quality of our manuscript. Please find below our reply to the individual points, marked with an "AC" (author's comment).

Best regards, Susanna Mohr on behalf of all co-authors

Response to the referee comments: Referee #1:

Separating the occurrences and measurements of different Aeolian phenomena such as synoptic cyclones, thunderstorms, tornadoes and so on is a key topic of modern wind engineering in order to perform distinct statistical analysis, to extract the main

statistical parameters related to each phenomenon, and to build wind field models suitable to represent the wind loading and response of structures. Merging these separate evaluations in a unitary formulation is a further aim still in the embryonic stage.

This paper provides very interesting and new information on several aspects in the above framework, thus represents a useful and pertinent contribution to the advance of the knowledge in this field. In its whole I appreciate it and support its publication.

This paper contains a broad literature both in the fields of atmospheric sciences and wind engineering, perhaps a little biased towards the first field. Despite this I believe that some relevant contributions to this topic are not considered and some choices inherent the methods herein applied seem to be based on a limited view of some previous contributions. Under this point of view, without changing anything in the substance of this paper, I believe that a wider critical discussion on the advantages and shortcomings related to such choices may improve the quality of this paper and inspire future step forwards.

More in detail, I recommend Authors to take into account the following remarks and observations:

Section 1: At least two additional references should be considered. The first (Gomes and Vickery,1978) is the fundamental paper that in 1977 introduced the concept of mixed wind climate and the idea of processing separately the statistical analysis of different wind phenomena. The second (Kasperski, 2002) published in 2002, deals with the same topic of the present paper just with reference to Germany. A comparison with previous methods and results is recommended.

AC: We will add both literatures in the introductions at the corresponding passages.

Section 2: I am quite doubtful on the decision of restricting analyses to the summer half-year. In my experience thunderstorm events are concentrated in this part of the year but are present also, in minor proportion, all over the year. Restricting analyses to a period

is even more dangerous considering the aim of performing a statistical analysis of the extreme wind speed. Unavoidably this produces underestimated results. I suggest to revise this choice in next contributions.

AC: In Germany, thunderstorms do not occur very often during the inter half year and, when they occur, those events are in general embedded in frontal systems, which is not our interest in this work (avoid mixed climate). For example, Wapler (2013) shows that in Germany the number of strokes during the winter half year is a power of 10 to 100 smaller as during the summer time. Further, she demonstrated exemplary for a few weather stations an extremely small number of thunderstorm days during the winter half season (< 1 per month). However, in future work we will investigate the sensitivity on the results considering the winter events more accurately.

Section 2.1: Authors base their analyses on the daily peak and subsequent mean wind speeds on 10-min and 1-h periods. They also use pressure measurements. A very similar approach is used in Uruguay and described in Duranona (2015). I suggest to examine this contribution.

AC: Thanks for the reference of this work. However, the author uses another definition to separate strong convective wind events from a mixed wind climate (sudden increases in wind speed, temperature drops, wind direction shifts) and they did not used pressure measurements/gradients. And the overlap between the studies concerns only the results regarding the (convective) gust factors. Therefore, we will incorporate the proposed literature only in the result section about the gust factor.

Section 2.1: Also in the light of the occurrence of gust factors in the order of 6-10, I suggest Authors to consider the possibility that some peak values in the database may be wrong (Cook, 2014). The potential presence of some mistakes and the difficulty of recognizing them is a major shortcoming of this kind of analyses, where the control is very good in terms on mean values but almost impossible with reference to single peaks.

AC: Again, we explicitly checked each event with high values above 6 and could not identify some mistakes. For example, in those cases the mean hourly wind are < 6 m/s. Choi and Hidayat (2002) have already stated that gust factors obtained close to the storm center may reach values between 7 and 8. Furthermore, those high values can certainly result when the event duration of a gust is very small or the event happens in the end of the measuring time and, thus, the gust affects not so much the hourly mean wind.

Section 2.3: I understand that Authors have probably no other opportunity than this use of lightning data. In my experience the presence of cloud-to-cloud lightning not detected by measurements may provide some relevant drawback. I verified this by comparing similar lightning data with high-sampling velocity records.

AC: That's right and we are also aware of this. But unfortunately cloud-to-cloud lightning (CC) was not recorded by BLIDS entirely due to the lower frequency range. However, studies show that thunderstorms connected with only CC occur predominantly during the winter time or are in general "weaker" (cf., Rakov and Uman, 2003) and, thus, less associated with strong downdrafts or straight-line winds. We will add a comment about this aspect/uncertainty in this section.

Section 2.4: The problem of the separation of different wind events is a key topic because any mistake in this stage may compromise the quality of further evaluations. I suggest Authors to dedicate a few more words to this problem for instance using a citation to Lombardo et al. (2009) (included in references but not cited here) and to De Gaentano et al. (2013).

AC: We will add more comments about this problematic in the section "Definition of convective gusts" considering the mentioned studies.

Section 2.5: Authors speak of GEV and POT/GPD and make the choice of using POT/GPD. This is fine but again, without changing the substance of this paper, this topic is a "world" that may necessitate a some more "delicate" approach. First of all the

use of POT/GPD is widely supported by some Authors but drastically opposed by others. The reference (Harris, 2005), for instance, is fully devoted to demonstrate that this method is wrong or at least unreliable. Our research group recently published a series of papers based on long-term Monte Carlo simulations (Torrielli et al., 2013; 2014) that confirms the limited reliability of the POT/GPD technique and arrives to the conclusion that the Process Analysis (Gomez and Vickery, 1977) (probably not easy to apply to thunderstorms) and the Penultimate distribution (Cook and Harris, 2004; 2008) are the best methods.

AC: We will address this issue and the related uncertainties (and literature) more in detail in the conclusion.   Section 2.5: At the end of this section Authors write "that the differences between the return values estimated by both methods are considerably smaller than the uncertainties of the method itself". This is absolutely correct with reference to return periods in the order of the number of years of available data, for instance 20-50 years. Structural safety, however, needs evaluations extrapolated to return periods in the order of 500-1000 years. Here, different methods lead to divergent results (Cook and Harris, 2004; 2008; Lagomarsino et al., 1992).

AC: Regarding the first comment, that's correct and we will specify that. Regarding the structural safety: In general, the reference velocity in national standards like DIN or EUROCODE is based on average on a return value of 50-year.

Section 3.1: Authors write: "we considered every single measurement at each station, which means that one event can be recorded on two or more stations". I think that this sentence may result misleading. Downbursts are phenomenon with a radius of a few km. It is almost impossible that the same downburst may be detected by two stations of this network. The situation is different if Authors refer to the large scale wind event that generates downbursts. This point should be clarified.

AC: We will specify that in the corresponding passage.

Section 3.3: The last sentence deserves a citation to Authors that first expressed this

concept (Gomes and Vickery,1978).

AC: We will include the literature at this point.

Section 3.4: Line 11. The dependence of the gust factor on the averaging period is discussed also by Solari et al. (2015).

AC: We will also refer the literature at this passage.

Section 3.4: Line 20. I do not agree on the sentence according to which "turbulent factors (to replace with gust factors ?) fluctuate usually between 1.2 and 2.3". Such a large variability necessarily depends not only on the roughness length but even more on the stability conditions. If wind is intense and of synoptic type, then atmosphere is neutrally stratified and the gust factor may vary between 1.25 and 1.75 with an average value around 1.5. In my opinion gust factors in the order of 1.75-2.3 may be ascribed to unstable conditions and intermediate events between large scale depressions and mesoscale downbursts (De Gaetano et al., 2013).

AC: We will correct that and will integrate the dependency of the gust factor values regarding to the atmospheric stability conditions.

Independently of the above remarks, that I wrote in a fully constructive spirit, I confirm my appreciation towards this contribution and that I consider it appropriate for publication. I hope that Authors may consider or discuss my remarks. I suggest that this paper is accepted subject to minor revisions but I believe that a quick re-review may be useful.

Giovanni Solari Department of Civil, Chemical and Environmental Engineering Polytechnic School, University of Genoa, Italy

  References: 1. Gomes L, Vickery BJ (1978). Extreme wind speeds in mixed climates. J Ind Aerod, 2, 331-344. 2. Kasperski M (2002). A new wind zone map of Germany. J Wind Eng Ind Aerod, 90, 1271-1287. 3. Duranona V (2015). The significance of non-synoptic winds in the extreme wind climate of Uruguay. Proc 14th Int

Conf on Wind Engineering, Porto Alegre, Brasil. 4. Cook NJ (2014) Review of errors in archived wind data. Weather 69: 72-78. 5. De Gaetano P, Repetto MP, Repetto T, Solari G (2013). Separation and classification of extreme wind events from anemometric data. J Wind Eng Ind Aerod, 126, 132-143. 6. Harris, R.I., 2005. Generalized Pareto methods for wind extremes. Useful tool or mathematical mirage? JWEIA 93, 897-918. 7. Torrielli A, Repetto MP, Solari G (2013). Extreme wind speeds from long-term synthetic records, J Wind Eng Ind Aerod, 115, 22-38. 8. Torrielli A, Repetto MP, Solari G (2014). A refined analysis and simulation of the wind speed macro-meteorological components. J Wind Eng Ind Aerod, 132, 54-65. 9. Gomes, L., Vickery, B.J., 1977. On the prediction of extreme wind speeds from the parent distribution. J. Ind. Aerodyn. 2, 21-36. 10. Cook, J., Harris, I., 2004. Exact and general FT1 penultimate distributions of extreme wind speeds drawn from tail-equivalent Weibull parents. Struct. Saf. 26, 391-420. 11. Cook, J., Harris, I., 2008. Postscript to "Exact and general FT1 penultimate distributions of extreme wind speeds drawn from tail-equivalent Weibull parents". Struct. Saf 30, 1-10. 12. Lagomarsino, S., Piccardo, G., Solari G., 1992. Statistical analysis of high return period wind speeds, JWEIA 41, 485-496.

AC: Thanks for the many literature suggestions. In the meantime, we have a rather comprehensive literature database, however, with a focus more on convective wind gusts and not so detailed on the general perspective of strong wind events.  

Response to the referee comments: Referee #2 (Anonymous):

General comments

The manuscript "Statistical characteristics of convective wind gusts in Germany" written by Susanna Mohr et al. describes a methodology to identify and select convective wind gusts from station measurements at 110 stations within Germany. Characteristics regarding the seasonality as well as spatial variations over Germany are considered and rare convective gusts are characterised by means of extreme value statistics. Additionally, by comparing the convective gust measurements to mean winds, gust factors are

quantified. Generally, the study presents very relevant work and is an important contribution to the understanding of local small scale convective wind gusts. The manuscript is well written and the chosen methods to assess the statistical characteristics of convective gusts in general seem appropriate and well suited. However, I notice several minor flaws (which I listed below) in the methodological and statistical approach which I would recommend the authors to consider. I thus suggest the paper to be accepted after minor revisions.

Specific comments

P. 3, L. 17: Results show, that no significant differences are found in the intensity of rare convective gusts with respect to orography. Why are stations at higher ground excluded? It might be particularly worthwhile to also consider stations at higher altitudes!

AC: At higher stations the separation and classification of strong wind events into homogeneous families (convectively driven) is very difficult and we explicitly want to exclude events caused by a mixed climate (large-scale / convective conditions). We will add a comment about this.

P. 4, L. 13-14: The choice of a 50-km radius does not seem to be justified by the given explanation. Since a gust front can occur several kilometers ahead of a storm center this might suggest a radius of 5, possibly 10 km.

AC: A gust front can occur several tens of kilometers ahead of a storm center with lightning activity, as already shown by Klingle et al. (1987) and Pantillion et al. (2015). Note that we consider only clout-to-ground lightning. We also refer to the concept of a "trailing gust front" (typically south or southwest of the flank of a storm; see also Houze, 2014; chapter 9), where higher distance to the storm center are possible. By the way, we tested a varied radius and a distance reducing does not modify the overall results. We will add comments about that in the revised version.

Klingle D. L.; Smith D. R. & Wolfson M. M. Gust front characteristics as detected by

Doppler radar Mon. Weather Rev., 1987, 115, 905-918.

Pantillon, F.; Knippertz, P.; Marsham, J. H. & Birch, C. E. A parameterization of convective dust storms for models with mass-flux convection schemes J. Atmos. Sci., 2015, 72, 2545-2561.

Houze R. A. Cloud Dynamics (2nd Ed.) Elsevier Inc., Oxford, UK, 2014.

P. 4, L. 20 "proximity to the wind station": Pressure gradients are calculated by means of a small set of 6 climate stations. It should be explained how the pressure gradients "in proximity to the wind station" are determined and in how far it can be expected that small scale depressions can be captured (or why such small scale depressions are disregarded!).

AC: The six climate stations are located over Germany and the distance between the stations is always smaller than 250 km (mean 210 km). Means that greatest pressure gradient of the nearest station in dependency to the others five stations is investigated. Therefore, we should capture with the filter approach also small scale depressions. We will discuss that more in detail in the revised version.

P. 4, L. 20-22: This additional filter criterion seems a bit random/unsystematic. I suspect, that not only in April but also in autumn such weaker pressure gradients do occur. I would thus favor a more systematic treatment of seasonality. Also, this additional criterion might hinder the interpretation of spatial as well as the seasonal variance discussed later in the text.

AC: You are right. This additional criterion seems to be a bit random. However, we performed several comparisons using synoptic weather charts and found an additional criterion is necessary, but only in April, where large-scale storm over the North Sea (not so common in September) may affect the gust statistics in the North German Plain (> 52°N). We will specify this.

P. 4, L 23-24: Sensitivity of what? It should be specified in which respect the sensitivity

has been considered!

AC: Both the distance to a lightning recording and the thresholds of pressure gradients were identified by sensitivity and individual case studies by investigation of the impact of varying thresholds on the sample size and the following results. We will add a comment.

P. 5, L. 2: What is meant by "as the approach reproduced the sample better"?

AC: We mean "as that approach reproduced the observed gusts better". We will change the formulation.

P. 5, L 3-4: Does "uncertainties of the method itself" refer to confidence intervals on estimated return values? Should be clarified!

AC: Yes. We will clarify that.

P. 6, L. 19-26: In Figure 1, I would propose adding confidence intervals to indicate the uncertainties in the seasonal variations. As noted above, a single event can cause the peak in beginning of April. Without a proper estimation of uncertainties (confidence intervals) I would challenge the statistical robustness of the results presented here!

AC: We will include the confidence intervals in the figure. Probably, only in Figure 1a to show the uncertainties and not in Figure 1b, since it could be confusing with the North/South information. We have to test this.

P. 7, L. 7-9: Has this been tested explicitly or is this just an interpretation of the missing north-to-south gradient? Of course this could be explicitly done correlating orographic height of the station against percentile value?! This is also related to my previous comment on excluding stations at higher locations.

AC: Yes. We have not recognized any correlation between the percentile values and station heights. The correlation is in general < - 0.1. In addition, it is striking that stations above 500 m (16 station means 15%) have smaller values than lower located

stations. We will add a new Figure and more comments about that in the revised version.

P. 7, L. 27-28: Although I do not want to object to the threshold choice itself, I do want to mention that this is not how the parameter stability criterion should be interpreted! In the GPD framework, it can be inferred that if the distribution of values above a certain threshold (u_0) follows a GPD, then it follows a GPD above all thresholds higher than u_0 with a modified sigma. Shape and modified scale should thus be constant above (not near) the chosen threshold within confidence intervals! For details see Coles: An Introduction to Statistical Modeling of Extremes, 2001 p. 78/79.

AC: That's correct. We will modify our unhappily chosen formulation. However, this point has already taken into account in the investigations.   P. 8, L. 3-5: A comparison of the empirical estimates (95% percentile) and estimates from extreme value statistics might be interesting here. According to the numbers that are specified in lines 10-11 on page 7 we are then talking about a return period of about 1 year.

AC: A comparison between percentiles and RV1a show that values with a return period of 1 year corresponding to a 95 % to 98 % percentile–depending on the stations. We will add a sentence.

P. 8, L. 14-15: It should be clarified how the statistical uncertainty is calculated for a region. In the caption of figure 5 it is mentioned that it corresponds to the mean of 95% confidence levels. I do not see why and how this should compare to the standard deviation for different stations (regional variability)!

AC: We will modify the figure caption to clarify this.

P. 10, L. 24: As mentioned in my previous comment, it should be clarified if this has been explicitly tested or weather this is simply the interpretation of Figures 2 and 4 (which do not contain an explicit information on orographic height).

AC: Yes. See comments above. Depending on RV the correlations (to station height)

are between 0.2 – 0.3.

P. 10, L. 26: By definition, an event with a 20-year return period has a fixed occurrence frequency of 20 years! Please rewrite!

AC: Yes, that is correct. We will rewrite this in "A comparison of the 20-year return values of convective gusts with those of turbulent gusts demonstrates that the latter have higher return values."

Technical corrections P. 7, L. 7: slight variability instead of slightly variability.

AC: Thank you for close reading. We will correct that.

---

## Author Response (AR1)

Dear Referees,

thank you very much for your work and the useful and valuable comments how to improve the scientific quality of our manuscript. Please find below our reply to the individual points, marked with an "AC" (author's comment).

Best regards,
Susanna Mohr on behalf of all co-authors

**Response to the referee comments: Referee #1:**

Separating the occurrences and measurements of different Aeolian phenomena such as synoptic cyclones, thunderstorms, tornadoes and so on is a key topic of modern wind engineering in order to perform distinct statistical analysis, to extract the main statistical parameters related to each phenomenon, and to build wind field models suitable to represent the wind loading and response of structures. Merging these separate evaluations in a unitary formulation is a further aim still in the embryonic stage.

This paper provides very interesting and new information on several aspects in the above framework, thus represents a useful and pertinent contribution to the advance of the knowledge in this field. In its whole I appreciate it and support its publication.

This paper contains a broad literature both in the fields of atmospheric sciences and wind engineering, perhaps a little biased towards the first field. Despite this I believe that some relevant contributions to this topic are not considered and some choices inherent the methods herein applied seem to be based on a limited view of some previous contributions. Under this point of view, without changing anything in the substance of this paper, I believe that a wider critical discussion on the advantages and shortcomings related to such choices may improve the quality of this paper and inspire future step forwards.

More in detail, I recommend Authors to take into account the following remarks and observations:

Section 1: At least two additional references should be considered. The first (*Gomes and Vickery,1978*) is the fundamental paper that in 1977 introduced the concept of mixed wind climate and the idea of processing separately the statistical analysis of different wind phenomena. The second (*Kasperski, 2002*) published in 2002, deals with the same topic of the present paper just with reference to Germany. A comparison with previous methods and results is recommended.

AC: We added both literatures in the introduction.

Section 2: I am quite doubtful on the decision of restricting analyses to the summer half-year. In my experience thunderstorm events are concentrated in this part of the year but are present also, in minor proportion, all over the year. Restricting analyses to a period is even more dangerous considering the aim of performing a statistical analysis of the extreme wind speed. Unavoidably this produces underestimated results. I suggest to revise this choice in next contributions.

AC: In Germany, thunderstorms do not occur very often during the winter half year and, when they occur, those events are in general embedded in frontal systems, which is not our interest in this work (avoid mixed climate). For example, Wapler (2013) showed that in Germany the number of strokes during the winter half year is one to two orders of magnitude smaller compared to the summer time. Furthermore, she identified exemplarily for a few weather stations an extremely small number of thunderstorm days during the winter half season (< 1 per month). However, in future work we will investigate the sensitivity on the results by considering the winter events more accurately.

Section 2.1: Authors base their analyses on the daily peak and subsequent mean wind speeds on 10-min and 1-h periods. They also use pressure measurements. A very similar approach is used in Uruguay and described in *Duranona (2015)*. I suggest to examine this contribution.

AC: Thanks for the reference of this work. However, the author uses another definition to separate strong convective wind events from a mixed wind climate (sudden increases in wind speed, temperature drops, wind direction shifts). They did not use pressure measurements/gradients, and the overlap between the studies concerns only the results regarding the (convective) gust factors. Therefore, we added the proposed literature only in the result section.

Section 2.1: Also in the light of the occurrence of gust factors in the order of 6-10, I suggest Authors to consider the possibility that some peak values in the database may be wrong (*Cook, 2014*). The potential presence of some mistakes and the difficulty of recognizing them is a major shortcoming of this kind of analyses, where the control is very good in terms on mean values but almost impossible with reference to single peaks.

AC: Again, we explicitly checked each event with high values above 6 and could not identify any inaccuracy. For example, in those cases the mean hourly wind was < 6 m/s. Choi and Hidayat (2002) have already stated that gust factors obtained close to the storm's center may reach values between 7 and 8. Furthermore, those high values can certainly result when the duration of a gust is small or the event happens at the end of the measuring time and, thus, estimated hourly mean wind speed is almost insensitive to individual gusts..

Section 2.3: I understand that Authors have probably no other opportunity than this use of lightning data. In my experience the presence of cloud-to-cloud lightning not detected by measurements may provide some relevant drawback. I verified this by comparing similar lightning data with high-sampling velocity records.

AC: That's right and we are aware of this. But unfortunately cloud-to-cloud lightning (CC) was not recorded by BLIDS entirely due to the lower frequency range. However, several studies have shown that thunderstorms connected with only CC occur predominantly during the winter time or are in general "weaker" (cf., Rakov and Uman, 2003) and, thus, to a less degree associated with strong downdrafts or straight-line winds. We added a comment about this aspect/uncertainty in this section.

*"Although three-quarters of all lightning is CC (Rakov and Uman, 2003), we assume that severe convective storms with strong downdrafts and thus strong gusts at the surface produce a sufficient number of CG flashes."*

Section 2.4: The problem of the separation of different wind events is a key topic because any mistake in this stage may compromise the quality of further evaluations. I suggest Authors to dedicate a few more words to this problem for instance using a citation to *Lombardo et al. (2009)* (included in references but not cited here) and to *De Gaentano et al. (2013)*.

AC: We added a comment about this problem in the section "Definition of convective gusts" considering the mentioned studies:

*"The separation and classification of strong wind events into synoptically/frontal driven and convectively driven gusts is a key topic, because any inadequacies in this stage may severely affect subsequent statistics. Different approaches have been established that are either based on individual detailed inspections of the prevailing meteorological conditions (by using surface measurements, radar/satellite images,...) during the wind events or on systematic filtering using proxy data for thunderstorms such as lightning. A summary of the different methods to extract convective gusts can be found in De Gaetano et al. (2014).*
*We follow the approach of Choi and Hidayat (2002) and Cook et al. (2003) and considered wind gusts only in combination with a thunderstorm day."*

Section 2.5: Authors speak of GEV and POT/GPD and make the choice of using POT/GPD. This is fine but again, without changing the substance of this paper, this topic is a "world" that may necessitate a some more "delicate" approach. First of all the use of POT/GPD is widely supported by some Authors but drastically opposed by others. The reference (*Harris, 2005*), for instance, is fully devoted to demonstrate that this method is wrong or at least unreliable. Our research group recently published a series of papers based on long-term Monte Carlo simulations (*Torrielli et al., 2013; 2014*) that confirms the limited reliability of the POT/GPD technique and arrives to the conclusion that the Process Analysis (*Gomez and Vickery, 1977*) (probably not easy to apply to thunderstorms) and the Penultimate distribution (*Cook and Harris, 2004; 2008*) are the best methods.

AC: We addressed this issue in the conclusion:

*"Concerning extreme value statistics, the use of POT/GPD method is discussed controversially in the literature. Whereas the use is widely supported by some authors (e. g., Hosking and Wallis, 1987; Simiu and Heckert, 1996; Palutikof et al., 1999; Holmes and Moriarty, 1999; Simiu, 2007), Harris (2005) demonstrated that this method has grave defects and thus is unreliable. Therefore, over recent years new methods have been developed to overcome this problematic, which are based, for instance, on long-term Monte Carlo simulations (Torrielli et al., 2013) or the Penultimate distribution (Cook and Harris, 2004, 2008). These approaches for several cases may help to improve the statistics and to reduce underlying uncertainty."*

Section 2.5: At the end of this section Authors write "that the differences between the return values estimated by both methods are considerably smaller than the uncertainties of the method itself". This is absolutely correct with reference to return periods in the order of the number of years of available data, for instance 20-50 years. Structural safety, however, needs evaluations extrapolated to return periods in the order of 500-1000 years. Here, different methods lead to divergent results (*Cook and Harris, 2004; 2008; Lagomarsino et al., 1992*).

AC: Regarding the first comment we have specified:

*"…the differences between the return values as considered in this work (<100 years) estimated from both methods are considerably smaller than the uncertainties of the method itself…"*

Regarding structural safety: In general, the reference velocity in national standards like DIN or EUROCODE is based on average on a return period of 50-year.

Section 3.1: Authors write: "we considered every single measurement at each station, which means that one event can be recorded on two or more stations". I think that this sentence may result misleading. Downbursts are phenomenon with a radius of a few km. It is almost impossible that the same downburst may be detected by two stations of this network. The situation is different if Authors refer to the large scale wind event that generates downbursts. This point should be clarified.

AC: We specified that:

*"Note that we considered every single measurement at each station, which means that one event can be recorded on two or more stations if event is related to a meso-scale cold pool."*

 Section 3.3: The last sentence deserves a citation to Authors that first expressed this concept (*Gomes and Vickery,1978*).

AC: We included the literature at this point.

Section 3.4: Line 11. The dependence of the gust factor on the averaging period is discussed also by *Solari et al. (2015).*

AC: We also referred to the literature in this paragraph.

Section 3.4: Line 20. I do not agree on the sentence according to which "turbulent factors (to replace with gust factors ?) fluctuate usually between 1.2 and 2.3". Such a large variability

necessarily depends not only on the roughness length but even more on the stability conditions. If wind is intense and of synoptic type, then atmosphere is neutrally stratified and the gust factor may vary between 1.25 and 1.75 with an average value around 1.5. In my opinion gust factors in the order of 1.75-2.3 may be ascribed to unstable conditions and intermediate events between large scale depressions and mesoscale downbursts (*De Gaetano et al., 2013*).

AC: We corrected that and integrated the dependency of the gust factor values regarding atmospheric stability:

*"Depending on the land use, surface roughness, or atmospheric stability, respectively, turbulent gust factors fluctuate usually between 1.2 and 2.3."*

Independently of the above remarks, that I wrote in a fully constructive spirit, I confirm my appreciation towards this contribution and that I consider it appropriate for publication. I hope that Authors may consider or discuss my remarks. I suggest that this paper is accepted subject to minor revisions but I believe that a quick re-review may be useful.

Giovanni Solari Department of Civil, Chemical and Environmental Engineering Polytechnic School, University of Genoa, Italy

**Response to the referee comments: Referee #2 (Anonymous):**

*General comments*

The manuscript "Statistical characteristics of convective wind gusts in Germany" written by Susanna Mohr et al. describes a methodology to identify and select convective wind gusts from station measurements at 110 stations within Germany. Characteristics regarding the seasonality as well as spatial variations over Germany are considered and rare convective gusts are characterised by means of extreme value statistics. Additionally, by comparing the convective gust measurements to mean winds, gust factors are quantified.

Generally, the study presents very relevant work and is an important contribution to the understanding of local small scale convective wind gusts. The manuscript is well written and the chosen methods to assess the statistical characteristics of convective gusts in general seem appropriate and well suited. However, I notice several minor flaws (which I listed below) in the methodological and statistical approach which I would recommend the authors to consider. I thus suggest the paper to be accepted after minor revisions.

*Specific comments*

P. 3, L. 17: Results show, that no significant differences are found in the intensity of rare convective gusts with respect to orography. Why are stations at higher ground excluded? It might be particularly worthwhile to also consider stations at higher altitudes!

AC: At higher elevated stations, the separation and classification of strong wind events into homogeneous families (convectively driven) is very difficult and we explicitly want to exclude events caused by a mixed climate (large-scale / convective conditions). We added a comment about this.

P. 4, L. 13-14: The choice of a 50-km radius does not seem to be justified by the given explanation. Since a gust front can occur several kilometers ahead of a storm center this might suggest a radius of 5, possibly 10 km.

AC: A gust front can occur several tens of kilometers ahead of a storm's center, as already shown by Klingle et al. (1987) and Pantillon et al. (2015). Note that we consider only clout-to-ground lightning. We also refer to the concept of a "trailing gust front" (typically south or southwest of the flank of a storm; see also Houze, 2014; chapter 9), where the distance between the gust and the highest lightning density is largest..

By the way, we have tested different radii and found no strong impact on the overall results. We added a comment on that in the revised version.

*"The reason for the large detection radius is that gust fronts can occur several tens of kilometers ahead of a storm center with lightning activity, as already shown by Klingle et al. (1987), Houze (2014), and Pantillon et al. (2015). Note that a lower distance does not modify the overall results."*

Klingle D. L.; Smith D. R. & Wolfson M. M. Gust front characteristics as detected by Doppler radar Mon. Weather Rev., 1987, 115, 905-918.

Pantillon, F.; Knippertz, P.; Marsham, J. H. & Birch, C. E. A parameterization of convective dust storms for models with mass-flux convection schemes J. Atmos. Sci., 2015, 72, 2545-2561.

Houze R. A. Cloud Dynamics (2nd Ed.) Elsevier Inc., Oxford, UK, 2014.

P. 4, L. 20 "proximity to the wind station": Pressure gradients are calculated by means of a small set of 6 climate stations. It should be explained how the pressure gradients "in proximity to the wind station" are determined and in how far it can be expected that small scale depressions can be captured (or why such small scale depressions are disregarded!).

AC: The six climate stations are more or less evenly distributed over Germany and the distance between two neighboring stations is always less than 250 km (mean 210 km). Means that the

pressure gradients between the nearest pressure station to the gust event and the others five pressure stations (cf., Section 2.1) is considered. Therefore, our filter also captures smaller-scale depressions (such as, e.g., gale Lothar in 1999, which was the most severe, but spatially most limited storm over recent decades). A comment is added in the revised version.

*"In addition to wind speed, daily pressure records (KL data) were used to filter out turbulent gusts (see Sect. 2.4). For this, the pressure differences among six climate stations located over Germany were considered (Schleswig, Norderney, Hannover, Berlin-Tempelhof, Frankfurt/Main-Airport, Hof, and Augsburg). The distance between the stations is in all cases less than 250km to capture also small scale depressions."*

P. 4, L. 20-22: This additional filter criterion seems a bit random/unsystematic. I suspect, that not only in April but also in autumn such weaker pressure gradients do occur. I would thus favor a more systematic treatment of seasonality. Also, this additional criterion might hinder the interpretation of spatial as well as the seasonal variance discussed later in the text.

AC: That's right, this additional criterion seems to be a bit random and cumbersome. However, we performed several comparisons using synoptic weather charts and found that an additional criterion is necessary for the month of April, where large-scale storms over the North Sea (not so common in September) may affect the gust statistics in the North German Plain (> 52°N). We specified this.

*"Because storm events related to low pressure systems frequently occur in April on the north coast, we used for this month and north of 52°N (North German Plain) a filter that excludes days under the influence of weaker pressure gradients compared to the first case 5 (>2.5 Pa 1/km). This criterion was extensively tested by a comparison with synoptic weather charts."*

P. 4, L 23-24: Sensitivity of what? It should be specified in which respect the sensitivity has been considered!

AC: Both the distance to a lightning recording and the thresholds of pressure gradients were identified by sensitivity and individual case studies. In doing so, we investigated the impact of varying thresholds on the sample size and the results of the statistics. We added a comment.

P. 5, L. 2: What is meant by "as the approach reproduced the sample better"?

AC: We mean "as that approach reproduced the observed gusts better". We changed the formulation.

P. 5, L 3-4: Does "uncertainties of the method itself" refer to confidence intervals on estimated return values? Should be clarified!

AC: Yes. We clarified that:

*"…the method itself (here expressed by confidence bounds)…"*

P. 6, L. 19-26: In Figure 1, I would propose adding confidence intervals to indicate the uncertainties in the seasonal variations. As noted above, a single event can cause the peak in beginning of April. Without a proper estimation of uncertainties (confidence intervals) I would challenge the statistical robustness of the results presented here!

AC: We included the confidence intervals in Figure 1a to show the variability. However, not in Figure 1b, since this might be confusing concerning the North/South discrepancy shown there.

P. 7, L. 7-9: Has this been tested explicitly or is this just an interpretation of the missing north-to-south gradient? Of course this could be explicitly done correlating orographic height of the station against percentile value?! This is also related to my previous comment on excluding stations at higher locations.

AC: We did not find any correlation between the percentile values and station heights. The correlation coefficient (Spearman) is in general < - 0.1. In addition, it is striking that stations above

500 m (16 station means 15%) have lower values than stations at lower elevations. We added a new Figure that shows the insensitivity of the estimated gusts on the station heights in the revised version.

P. 7, L. 27-28: Although I do not want to object to the threshold choice itself, I do want to mention that this is not how the parameter stability criterion should be interpreted! In the GPD framework, it can be inferred that if the distribution of values above a certain threshold (u_0) follows a GPD, then it follows a GPD above all thresholds higher than u_0 with a modified sigma. Shape and modified scale should thus be constant above (not near) the chosen threshold within confidence intervals! For details see Coles: An Introduction to Statistical Modeling of Extremes, 2001 p. 78/79.

AC: We modified this formulation. However, this point has already been taken into account in the investigations.

P. 8, L. 3-5: A comparison of the empirical estimates (95% percentile) and estimates from extreme value statistics might be interesting here. According to the numbers that are specified in lines 10-11 on page 7 we are then talking about a return period of about 1 year.

AC: A comparison between percentiles and RV1a shows that values with a return period of 1 year correspond to 95 % to 98 % percentiles–depending on the stations. We added a sentence.

P. 8, L. 14-15: It should be clarified how the statistical uncertainty is calculated for a region. In the caption of figure 5 it is mentioned that it corresponds to the mean of 95% confidence levels. I do not see why and how this should compare to the standard deviation for different stations (regional variability)!

AC: We modified the figure caption to clarify this:

*"Mean return values of convective gusts (RV) for various return periods in four regions in Germany (gray area). Red lines indicate the standard deviation from all stations within the respective region, and black bars indicate the mean 95% confidence intervals* representing *statistical uncertainties of the POT/GPD method."*

P. 10, L. 24: As mentioned in my previous comment, it should be clarified if this has been explicitly tested or weather this is simply the interpretation of Figures 2 and 4 (which do not contain an explicit information on orographic height).

AC: Yes. See comments above. Depending on RV the correlations (to station height) are between 0.2 – 0.3.

P. 10, L. 26: By definition, an event with a 20-year return period has a fixed occurrence frequency of 20 years! Please rewrite!

AC: Rewritten:

*"A comparison of the 20-year return values of convective gusts with those of turbulent gusts demonstrates that the latter have higher return values."*

*Technical corrections*

P. 7, L. 7: slight variability instead of slightly variability.

AC: Corrected.

[revised manuscript text omitted]
 ($\text{RV}_{20a}$) and 50 years ($\text{RV}_{50a}$) are on average $27.8 \pm 2.5 \, \text{m s}^{-1}$ and $30.2 \pm 3.1 \, \text{m s}^{-1}$, respectively (Fig. 5). In addition, isolated high values are estimated between 32 and $36 \, \text{m s}^{-1}$ for $\text{RV}_{20a}$ and between 36 and $40 \, \text{m s}^{-1}$ for $\text{RV}_{50a}$. High return values of up to $50 \, \text{m s}^{-1}$ for $\text{RV}_{50a}$ as estimated by Lombardo (2012) for the West Texas region (United States) cannot be estimated for Germany. Note, however, that a value above $50 \, \text{m s}^{-1}$ has already been observed (cf. Fig. 2c, **Zinnwald-Georgenfeld), but with an extremely high estimated return period of approximately 1,000 years (see also Seregina et al., 2014, supplemental material Table 1)**. Depending on the respective probability density function (or the parameters $k$ and $\sigma$; see Fig. 4e and f), the increase between both return values substantially differs among the stations. Although the differences between $\text{RV}_{20a}$ and $\text{RV}_{50a}$ are on average $2.3 \pm 0.9 \, \text{m s}^{-1}$, there are individual stations with differences between 4 and $6 \, \text{m s}^{-1}$ (e.g., Lahr; blue dot in the southwest corner of Fig. 5b). Furthermore, a comparison with Figure 2b shows that stations with similar 95 % percentile values do not have to show similar return values influenced by the underlying probability density function as well. **For example, convective gusts with a one year return period ($\text{RV}_{1a}$) correspond—depending on the station—to its 95 % to 98 % percentiles (correlation of 0.97).**

Depending on the station and thus on the individual shape parameter $k$, the uncertainty due to the application of an extreme value statistic is in a range between 3 and $16 \, \text{m s}^{-1}$ with a median of $6.5 \, \text{m s}^{-1}$ for $\text{RV}_{20a}$ and between 4 and $25 \, \text{m s}^{-1}$ with a median of $9.5 \, \text{m s}^{-1}$ for $\text{
[revised manuscript text omitted]